# Sound vs. light: wing-based communication in Carboniferous insects

Thomas Schubnel [1,5✉], Frédéric Legendre [1,5], Patrick Roques[2], Romain Garrouste[1], Raphaël Cornette[1], Michel Perreau[3,4], Naïl Perreau[4], Laure Desutter-Grandcolas[1,5] & André Nel [1,5✉]

Acoustic communication is well-known in insects since the Mesozoic, but earlier evidence of this behavior is rare. Titanoptera, an 'orthopteroid' Permian-Triassic order, is one of the few candidates for Paleozoic intersex calling interactions: some specimens had highly specialized broadened zones on the forewings, which are currently considered—despite inconclusive evidence—as 'resonators' of a stridulatory apparatus. Here we argue that the stridulatory apparatus hypothesis is unlikely because the Titanoptera lack a stridulatory file on their bodies, legs or wings. Instead, comparing these broadened zones with similar structures in extant locusts, flies, and fossil damselflies, we find evidence that the Titanoptera used their wings to produce flashes of light and/or crepitated sounds. Moreover, we describe the first Carboniferous (~310 Mya) Titanoptera, which exhibits such specialized zones, thus corresponding to the oldest record of wing communication in insects. Whether these communication systems were used to attract sexual partners and/or escape predators remain to be demonstrated.

[1] Institut de Systématique, Évolution, Biodiversité (ISYEB), Muséum national d'Histoire naturelle, CNRS, SU, EPHE, UA, 57 rue Cuvier, Paris Cedex 05, France. [2] Allée des Myosotis, Neuilly sur Marne, France. [3] IUT Paris Diderot, Université de Paris, 20 quater rue du département, Paris, France. [4] 27 quai d'Anjou, Paris, France. [5] These authors contributed equally: Thomas Schubnel, Frédéric Legendre, Laure Desutter-Grandcolas, André Nel.
✉email: thomas.schubnel@wanadoo.fr; anel@mnhn.fr

In animals, communication mechanisms are among the most important factors in evolution. They are known in all Metazoan groups and take a great variety of forms[1]. In insects, communication frequently involves wings, viz. for the diffusion of pheromones as in Lepidoptera and Trichoptera[2,3], production of flashes of light as in several Lepidoptera and Diptera[4,5], or emission of sounds as in Orthoptera, with a large variety of communication designs in each order. The latter use wings, and legs in some cases, to produce sounds in order to escape predators, or/and for intraspecific recognition, territorial delimitation and sexual calls.

Whereas the Triassic record abounds with evidence of insect communication, especially for Orthoptera Ensifera[6,7], the Palaeozoic record is reduced. Only the archaeorthopteran Permostridulidae Béthoux et al.[8] (order Caloneurodea Martynov, 1938) and some undescribed Ensifera, both from the Middle Permian, had stridulatory files on their tegmina[8-Huangetalinprep], and, putatively, the Permian-Triassic Titanoptera Sharov, 1968[9,10], an order of probably carnivorous 'giant' insects (Supplementary Fig. 1), currently placed in the Archaeorthoptera Béthoux and Nel, 2002 (see Supplementary Discussion: Phylogenetic relationships of Titanoptera). The Titanoptera mainly diversified during the Triassic (Australia and Central Asia)[6] and were putatively recorded in the Permian of Russia[9,10] (but see remark in Supplementary Discussion: Antiquity of Titanoptera). These very large insects died out during the latest Triassic or at the Triassic-Jurassic boundary, when the smaller predatory Mantodea with similar grasping legs would have started their diversification[11].

Many Titanoptera have highly specialized forewings (tegmina) with unique broad zones in their mid-part. Since McKeown[12], these zones are regarded as resonators of 'stridulatory' apparatuses,

present in males but not in females[13,14]. Sharov[6] disagreed and indicated that males and females of *Gigatitan vulgaris* Sharov, 1968 had similar wings and were both able to stridulate. He generalised his assumption to the whole order without further evidence. Later papers on animal sounds in the deep past took for granted the presence of stridulatory structures in Titanoptera wings[15].

Here we report and describe *Theiatitan azari* Schubnel, Roques & Nel, n. gen., n. sp., the oldest Titanoptera, from a Late Carboniferous wing (Avion, North of France). We provide a review of wing-based communication in insects and, comparing *Theiatitan* and other fossils, we reassess the highly specialized structures found in titanopteran tegmina and critically discuss how those insects communicated.

## Results and discussion

**Systematic palaeontology.** Insecta Linné, 1758;
 Archaeorthoptera Béthoux & Nel, 2002;
 Titanoptera Sharov, 1968;
 Family Theiatitanidae Schubnel, Roques & Nel, fam. nov.
 urn:lsid:zoobank.org:act:7F541872-8916-45B0-8654-4E394E8A48CF.

**Type genus and species.** *Theiatitan azari* Schubnel, Roques & Nel, gen. et sp. nov.;
 Genus name is registered under urn:lsid:zoobank.org:act:0F6-BAADF-E735-4FE9-B450-A2DD214D26E4;
 Species name is registered under urn:lsid:zoobank.org:pub:5F33D135-D38F-4D10-ADF5-9CE9BBFDC4C4
 (Fig. 1, Supplementary Fig. 1).

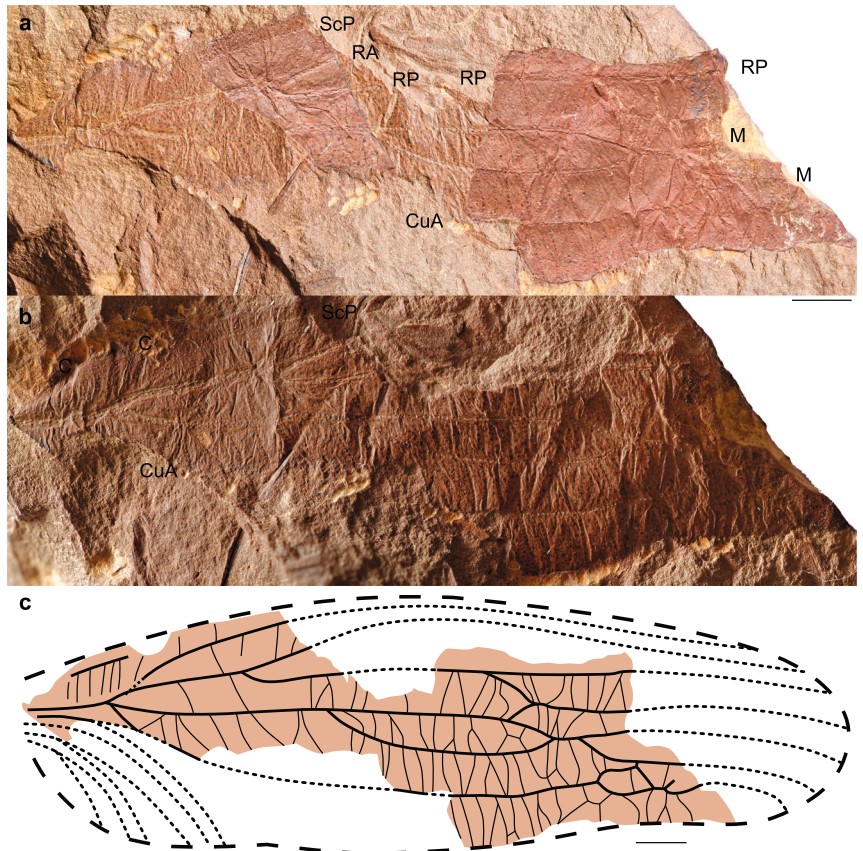

**Fig. 1 Oldest Carboniferous Titanoptera.** *Theiatitan azari* gen. & sp. nov., holotype MNHN.F.A70111. **a** Superposed view of imprint and counterimprint. **b** Basal part of imprint. **c** Reconstruction. C costa, ScP subcostal posterior, RA/RP radial anterior/posterior, M medial, CuA/CuP cubital anterior/posterior. Scale bars: 2 mm (**a**), 1 mm (**b–c**).

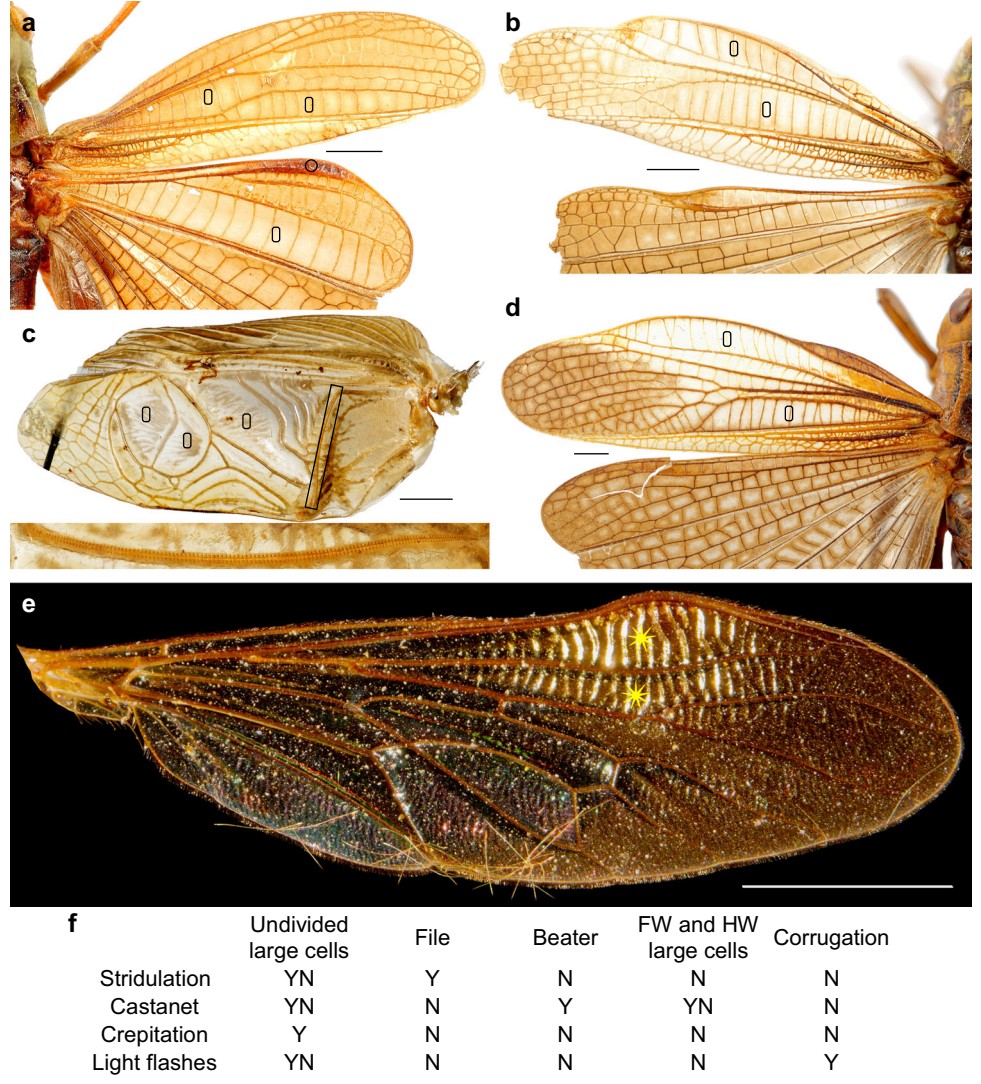

**Fig. 2 Structures in extant insects used to communicate.** (**a**, **b**, **d** Acrididae), (**c** Gryllidae), (**e** Asilidae). **a** *Stenobothrus rubicundulus* Kruseman & Jeekel, 1967, with enlarged cells (0) of castanet apparatuses on fore- and hind wings, and beater (°) on hind wing anterior margin. **b** *Stauroderus scalaris* (Fischer von Waldheim, 1846) male, enlarged cells (0) of crepitation zones. **c** *Gryllus bimaculatus* De Geer, 1773, male, stridulatory apparatus with enlarged cells (0) and file (narrow rectangle and detail below). **d** *Arcyptera fusca* (Pallas, 1773), enlarged cells (0) of crepitation zones. **e** *Ommatius torulosus* (Becker, 1925) male, corrugated reflecting zones (*). Scale bars: 2 mm. **f** Synthesis of the communication modes used in extant insects with their associated structures. Y yes, N no, YN co-occurrences.

| | Undivided large cells | File | Beater | FW and HW large cells | Corrugation |
|---|---|---|---|---|---|
| Stridulation | YN | Y | N | N | N |
| Castanet | YN | N | Y | YN | N |
| Crepitation | Y | N | N | N | N |
| Light flashes | YN | N | N | N | Y |

**Etymology**. The generic name refers to Theia, the Titanide of light in the Greek mythology, while 'titan' refers to the common suffix of the Titanoptera. The gender of the name is masculine. The specific epithet refers to our friend and colleague Pr. Dany Azar.

**Material**. Holotype MNHN.F.A70111 (Avion 37), sex unknown, imprint and counterimprint of mid part of a wing, collected by Patrick Roques; MNHN, Paris, France.

**Locality and horizon**. 'Terril N 7', containing rocks from the slag heap of coal mines 3 and 4 of Liévin, Avion, Pas-de-Calais, France; Moscovian (Westphalian C/D or equivalent Bolsovian/ Asturian), Carboniferous.

**Diagnosis**. Forewing venation characters only (Fig. 1a–c). Wing tegminized, with aligned small spines on the dorsal side of longitudinal veins (Fig. 1b); main veins not S-shaped; RP long and straight; broad zones between RP and M, branches of M, M

and CuA, and between CuA and posterior wing margin; numerous concave veinlets perpendicular to main veins in these zones, separating cells each with a convex surface; part of CuA basal of its fusion with CuPaα long; distal part of CuA(+CuPaα) very long and distally parallel to posterior wing margin; free branch(es) of CuPaα and CuPaβ short, not reaching mid part of wing (not present in this part of wing).

**Description and discussion on the affinities**. See Supplementary Discussion: Taxonomy.

**Wing-based communication in insects**. Various insects use their wings to communicate through air-borne physical signals or light signals (we do not discuss chemical signals, nor vibrations through the substrate[16]). Sounds can result from stridulation, crepitation or castanet mechanisms[17–19], and light flashes are produced through passive reflection on specialized wing areas. We present here those ways of signaling and associated structures before describing wing structures in Titanoptera. We then argue

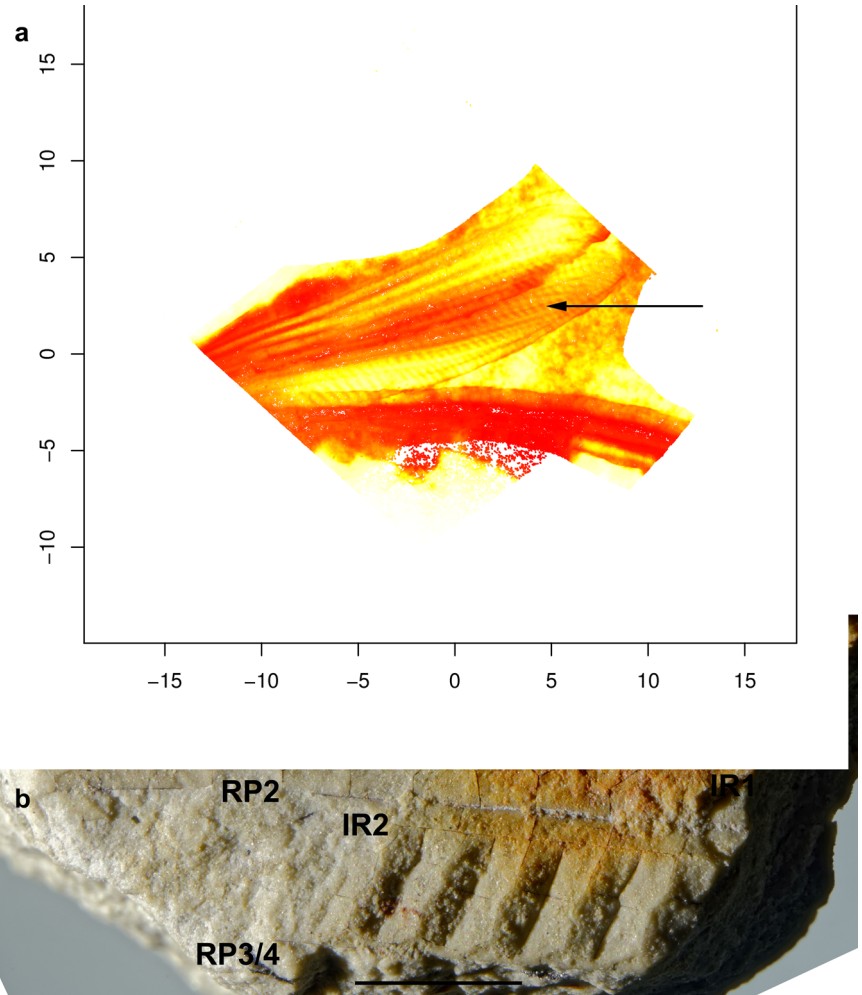

**Fig. 3 Odonata Steleopteridae with specialized zones probably producing light flashes. a** *Parasteleopteron guischardi* Fleck et al.[33] (Jurassic), holotype SOS 3615. False colors showing the relative reliefs of surfaces; arrow specialized zone; deep zones in red, higher zones in yellow. **b** *Steleopteron cretacicus* Zheng et al.[34] (Lower Cretaceous), showing differences of luminosity of parallel surfaces in specialized zone. RP2, RP3/4 branches of radius posterior, IR2 longitudinal intercalary vein. Scale bar: 2 mm.

that none of the stridulation, crepitation, castanet signaling or light flash alone fully explains the diversity of structures observed in Titanoptera.

*Stridulation.* Stridulation is defined as the production of sounds by rubbing one part of the body against another. In crickets and katydids, sound production involves stridulatory apparatuses comprising a file on one wing, a plectrum on the other; this design can be completed by a resonator called either the harp (present and symmetrical on both forewings) in crickets (Fig. 2c, f), or the mirror (asymmetrical or on left forewing only) in katydids. During stridulation, the plectrum is rubbed against the file in a lateral to-and-through movement of the raised tegmina. This movement generates a low-frequency sound; the resonator multiplies its frequency and amplifies it[20]. In some Orthoptera, the forewing is rubbed against another part of the body: hind legs, pronotum or hind wings. In these cases, the resonator is lacking, and the file is either on the wing or on the other involved body part. In species using their hind legs, e.g., Orthoptera Tropidopolinae (Supplementary Fig. 2), Gomphocerinae or Oedipodinae or some Lepidoptera Noctuidae[21], stridulation is possible only because at least the lateral part of the wing is oriented vertically to the body.

*Crepitation.* Crepitation is defined as the production of sounds by a weak wing membrane between veins that suddenly expands and vibrates due to a fast airflow[22]. Some extant Acridinae, Gomphocerinae, and Oedipodinae grasshoppers crepitate with fenestrated zones of their hind wings when they fly away, even for a short distance[19,23,24]. Some other grasshoppers (e.g. *Rhaphotittha* Karsch, 1896, *Stauroderus scalaris* (Fischer von Waldheim, 1846), *Arcyptera fusca* (Pallas, 1773) and *Chorthippus* Fieber, 1852) have similar crepitating structures in their forewings, more developed and better organized in males than in females (Fig. 2b, c, f). The males of the katydid *Segestidea queenslandica* possibly crepitate (see Supplementary Notes: Possible crepitation in a Tettigoniidae). All these insects use crepitation in male–male and male–female interactions, but also as escape behaviors. Whatever its function, crepitation is only possible with the combination of a thin elastic membrane and large cells. If crepitation can also occur when the animal is at rest, by quick movements of the forewings[25], it is very rare and most of the time, insects that produce sounds through crepitation do so only when they fly.

*Castanet.* Castanet is a design that produces sounds by a beat and clash of wings[26]. It can be done with one or two pairs of wings. Examples of castanet sounds made with only one pair of wings

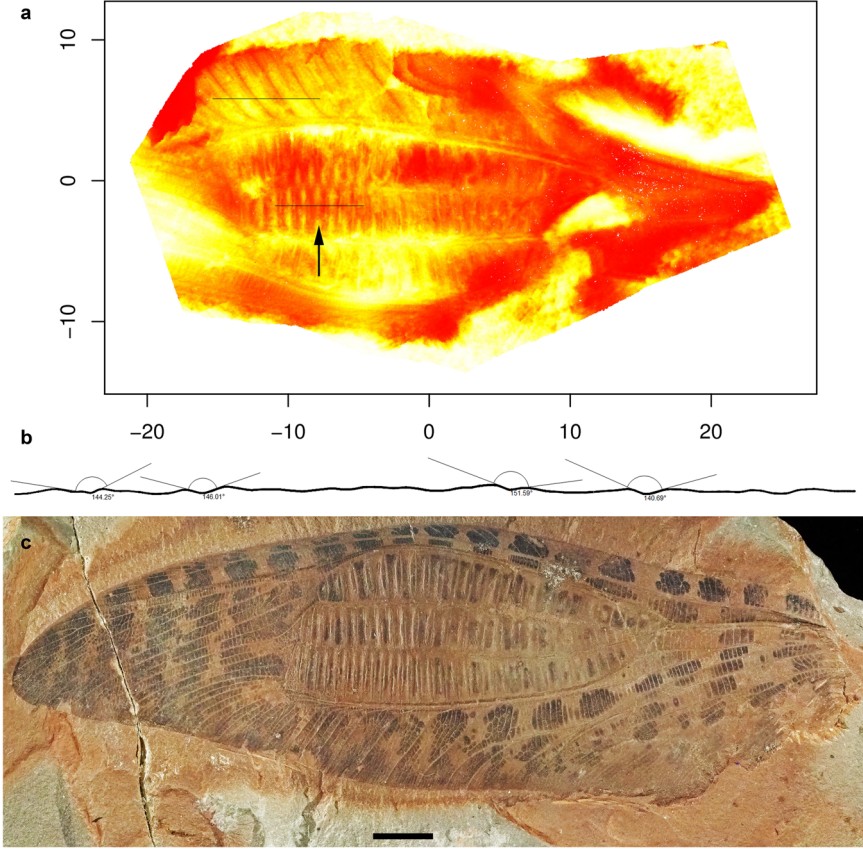

**Fig. 4 *Clatrotitan andersoni* McKeown[12], likely producing flashes of light.** Specimen NHM In. 37341. **a** False colors showing the relative reliefs of surfaces; line above: unspecialized zone; line below and arrow: specialized zone; deep zones in red, higher zones in yellow. **b** Virtual cross-section of specialized zone showing the angles between the different surfaces along line below. **c** AMF36274. Forewing. Scale bar: 10 mm.

are known in some Lepidoptera Noctuidae that use a cuticular knob surrounded by a pleated band of cuticle on the anterior edge of the forewings and clap their forewings together above the thorax[26]. In extant orthopteroids, a castanet apparatus implies enlarged cells on the two pairs of wings. A few Acrididae (e.g., *Stenobothrus rubicundulus* Kruseman & Jeekel, 1967 or *Fenestra bohlsii* Giglio-Tos, 1895) produce sounds with this mechanism, using the enlarged areas with large transverse cells of their fore- and hindwings, plus a beater (narrow and strongly sclerotized area between RA and RP) on the anterior margin of the hind wing (Fig. 2a, f).

*Light flashes.* Light flashes can be used to communicate between conspecifics or to confuse predators, as in the Lepidoptera Nymphalidae Morphinae or in other insects with metallic, structural colors[27–29]. Several insects are known to produce light flashes when flying, as sexual signals: some Acrididae[19]; damselflies of the family Chlorocyphidae Cowley, 1937, that have dark wings with several hyaline 'windows'; Ithomiini nymphalid butterflies with transparent wings; males of the calliphorid fly *Lucilia sericata* (Meigen, 1826)[5]; or males (but not females) of the asilid flies *Ommatius* spp., with broadened and transverse 'corrugated' cell r1[30,31] (Fig. 2e). The exact function of this broadened cell remains unknown, but as it is only in males and not in females, it is likely used for sexual communication (including territory delimitation). Many other insects have corrugated zones in the distal part of their wings (e.g. Hymenoptera Scoliidae or Siricidae) but these zones are identical in both sexes, less salient, and possibly serve to provide rigidity to transverse bending[32].

Last but not least, the damselflies of the Mesozoic family Steleopteridae Handlirsch, 1906 have highly modified groups of cells showing concave and convex veinlets defining surfaces of different orientations between their main longitudinal veins[33,34] (Fig. 3a–b). These extinct Zygoptera probably did not produce sounds of any kind with these structures, which remained extended away from the body and were not overlapping. No sound communication or hearing sense is anyway known in the crown group of Odonata[35], or documented in the large fossil record for Odonatoptera. More likely, they could have produced flashes of light during their flight. Overall, light flashes can be produced in two main ways: (1) because of a color variation in the wing; (2) by a corrugated structure that reflects light under specific angles (Fig. 2f).

**Specialized broadened structures in Titanoptera wing.** Most of the Titanoptera have broadened areas on the forewings, sometimes named 'speculum'[36], although they are narrower in some species than in others (e.g. *Prototitan sharovi* Gorochov[36] vs. *Clatrotitan andersoni* McKeown[12]). These zones can encompass several vein fields from the posterior branch of the radius vein (RP) to the posterior branch of the cubitus vein (CuP) and posterior wing margin (Figs. 4, 5; Supplementary Discussion: Distribution of broadened zones on tegmina of Titanoptera). They are, however, always present at least between the media vein (M) and the anterior branch of the cubitus vein (CuA), suggesting they are homologous across Titanoptera. These broadened zones contain large transverse cells, subdivided in some cases into a net of smaller cells, more or less regular[37] (see Figs. 4, 5a–d, Supplementary Fig. 3 and Supplementary Information: Reflection of

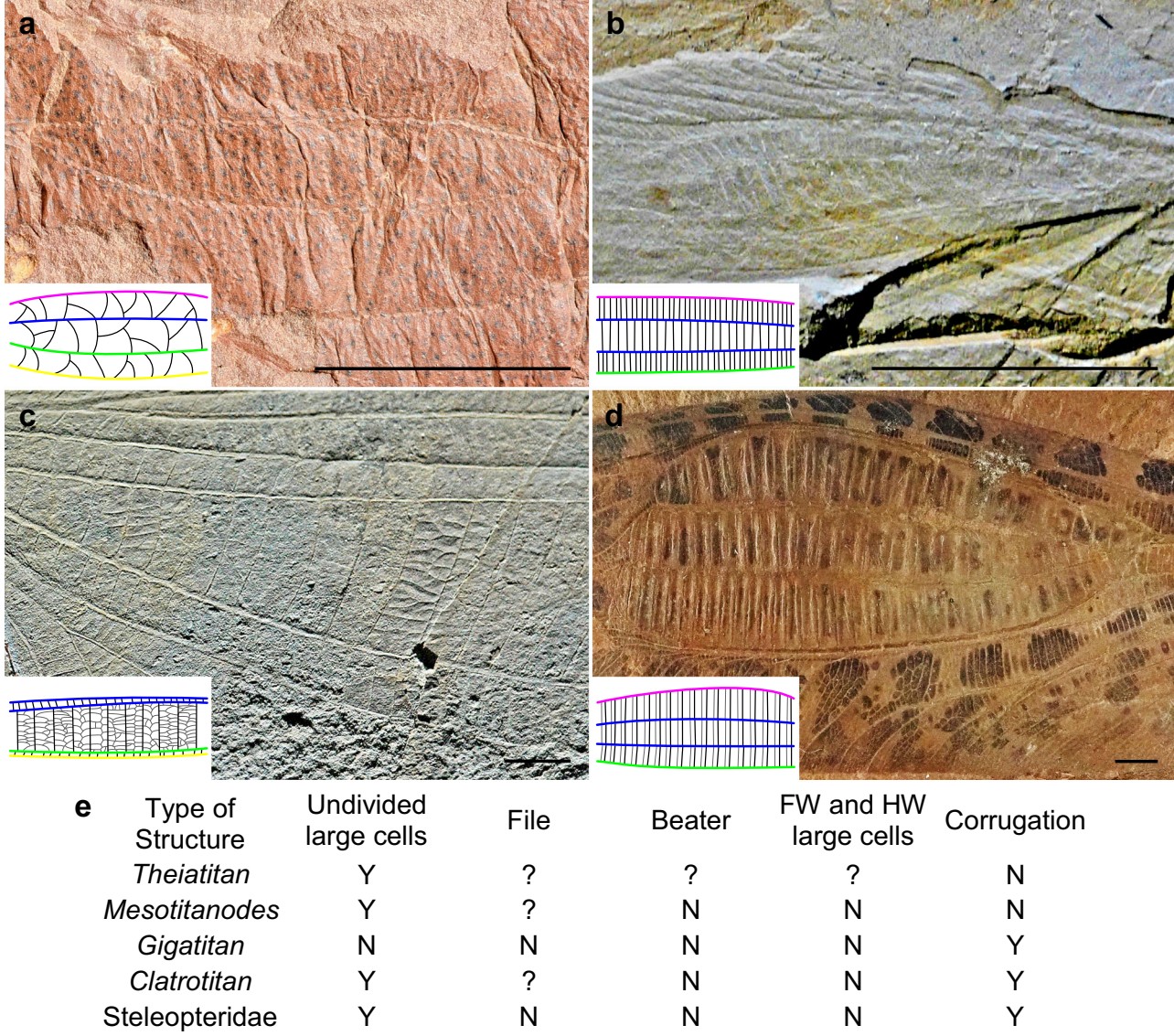

| e Type of Structure | Undivided large cells | File | Beater | FW and HW large cells | Corrugation |
|---|---|---|---|---|---|
| *Theiatitan* | Y | ? | ? | ? | N |
| *Mesotitanodes* | Y | ? | N | N | N |
| *Gigatitan* | N | N | N | N | Y |
| *Clatrotitan* | Y | ? | N | N | Y |
| Steleopteridae | Y | N | N | N | Y |

**Fig. 5 Diversity of broadened specialized areas in Titanoptera. a** *Theiatitan*, areas between RP, M, CuA and CuP, series of relatively irregular intercalary veinlets separating large cells. **b** *Mesotitanodes*, area between branches of M with regular large cells. **c** *Gigatitan*, area between branches of M with regular large cells subdivided into irregular nets of small cells. **d** *Clatrotitan*, areas between RP, branches of M, and CuA, with highly specialized cells. Scale bars: 5 mm. **e** Distribution of the structures used to communicate (see Fig. 2 for more details) among the Titanoptera. Colors of veins: pink radial vein; blue median vein; green cubitus anterior vein; yellow cubitus posterior.

light). In *Clatrotitan andersoni* and some others, the specialized zones are subdivided into a regular series of straight veinlets alternatively concave and convex (Fig. 4a) defining couples of flat cells with different oblique orientations (Fig. 4b). In each cell couple, the basal cell is declining, hyaline and smaller than the apical cell, which is ascending and darkened (Fig. 4c)[37]. The size ratio and the angle between the two cells of each couple are quite constant (respectively 2:1 and 146° ± 6), and the juxtaposition of these cell couples form a regular, corrugated structure.

**Oldest known Titanoptera—*Theiatitan azari* Schubnel, Roques & Nel, gen. et sp. nov.** *Theiatitan azari* Schubnel, Roques & Nel, gen. et sp. nov. is here described from a Lagerstätte dated from the late Carboniferous (310 Ma); *Theiatian* is thus 50 Ma older than the previous oldest Titanoptera. *Theiatitan* is unique among the Titanoptera in the CuA and CuP/wing margin broadened areas and veins displaying small spines, which support its

attribution to the new family Theiatitanidae Schubnel, Roques & Nel, fam. nov. *Theiatitan* also possesses the more classical M broadened areas, which is supposed to be used for communication. Thus, *Theiatitan* would be the oldest known insect with a wing structure specialized for communication.

**Flight of Titanoptera**. As we saw earlier, communication is often linked with flight. To test this relation in Titanoptera, we tried to study their flight ability. Several Titanoptera have long and broad wings that would have allowed them to fly. It is the case of *Gigatitan vulgaris*, one of the few Titanoptera for which both fore- and hind wings and body are known (see Supplementary Fig. 4). To study its flight ability, we measured the hind wing area of 22 modern Orthoptera with large wings (i.e. able to fly) and compared this to their body volume estimated as follows: volume = (width of thorax)$^2$ × body length (supposing the body is cylindrical, with a weak variation of width between thorax and

**Table 1 Wing surface and body volume of extant flying Orthoptera and *Gigatitan vulgaris*. Without indication, used specimens were males.**

|  | Length (mm) | Width (mm) | Volume (mm³) | Hind wing area (mm²) |
|---|---|---|---|---|
| *Platycleis albopunctata* | 16.3 | 1.96 | 62.61 | 97.09 |
| *Phaneroptera falcata* | 13.5 | 1.82 | 44.71 | 283.1 |
| *Ruspolia nitidula* | 20.39 | 2.65 | 143.18 | 182.19 |
| *Sanaa imperialis* | 30.05 | 4.96 | 739.27 | 764.85 |
| *Steirodon* sp. | 35.42 | 5.81 | 1195.64 | 1402.33 |
| *Clonia wahlbergi* | 51.02 | 2.76 | 388.64 | 1603.9 |
| *Siliquofera grandis* | 59.2 | 7 | 2900.8 | 2908.26 |
| *Pseudophyllanax imperialis* (female) | 61.37 | 12.75 | 9976.46 | 5338.62 |
| *Stenobothrus rubicundulus* | 17.93 | 2.07 | 76.82 | 95.09 |
| *Stauroderus scalaris* | 18.3 | 1.74 | 55.40 | 114.61 |
| *Oedipoda caerulescens* | 16.79 | 2.51 | 105.77 | 150.66 |
| *Arcyptera fusca* | 28.26 | 3.04 | 261.16 | 212.1 |
| *Oedalus decorus* | 31.25 | 2.57 | 206.40 | 313.73 |
| *Locusta migratoria* | 32.84 | 2.93 | 281.92 | 467.6 |
| *Phymateus saxosus* | 43.76 | 3.81 | 635.22 | 647.72 |
| *Anacridium aegyptium* | 46.91 | 4.46 | 933.11 | 867.98 |
| *Schistocerca gregeria* | 50.3 | 4.11 | 849.67 | 857.55 |
| *Titanacris picticrus* | 44.76 | 3.74 | 626.08 | 812.43 |
| *"Xiphocera cinerascens"* | 54.51 | 6.21 | 2102.12 | 1224.5 |
| *Tropidacris cristata* | 61.8 | 4.37 | 1180.18 | 1607.99 |
| *Titanacris picticrus* (female) | 64.36 | 5.46 | 1918.67 | 1473.58 |
| *Tropidacris cristata* (female) | 91.55 | 6.82 | 4258.21 | 2853.7 |
| *Gigatitan vulgaris* | 97.62 | 12.33 | 14841.06 | 5251.79 |

abdomen, and minimal deformation of the width of the thorax compared to the less sclerotized abdomen in the fossil compressions). Then we compared these results with the estimated body volume and hind wing area of *Gigatitan vulgaris*. We observed a high correlation between the wing surfaces and body volumes (Table 1, Fig. 6) for modern Orthoptera ($R^2 > 0.9$).

The hind wing area of *Gigatitan vulgaris* is almost the same as that of *Pseudophyllanax imperialis*, one of the largest modern Orthoptera. However, the estimated body volume of *G. vulgaris* is around 150% that of *P. imperialis*. As the latter has poor flying abilities, it seems highly improbable that *G. vulgaris* was able to actively fly. Passive gliding, cannot however be excluded for such large Titanoptera, even if good gliders have frequently hind wings distinctly broader and larger than forewings[38,39]. Also, as it is not possible to test the flight abilities of most Titanoptera, it cannot be excluded a priori that smaller Titanoptera with large wings could have been able to actively fly. But all known hind wings of Titanoptera, whatever their sizes, have quite reduced vannus[6], while most extant flying Orthoptera have large ones (see Fig. 6d).

**Nature of wing signaling in Titanoptera.** The particular broadened zones of the titanopteran forewings have been originally regarded as resonators of stridulatory apparatuses[12]. But, in the same work, McKeown[12] also indicated that 'in the absence of the body of the insect, it is not apparent how any sound could have been produced'. Despite this essential note, this hypothesis has been accepted without further investigations, even though neither the structure nor the functioning of this 'stridulatory' apparatus has been clearly settled so far.

The expanded alleged 'resonator' surfaces vary greatly in size and position in Titanoptera. Only *Theiatitan* and *Minititan* Gorochov[10] (replacement name for *Microtitan* Gorochov[36]) have cells as large as the resonator of some Ensifera; in other Titanoptera, the wing structures differ from this pattern. Further, no stridulatory file has ever been found on titanopteran insects; whether they were able to stridulate with a file—that has never been observed—a plectrum and a resonator, seems unlikely,

especially because an ensifera-like stridulation implies raising the wings and moving them to-and-through above the body.

An alternative would be that the titanopteran 'resonator' structures themselves were stridulatory files. This could seem congruent with the small spines present in the veins of *Theiatitan*, but the friction of the veinlets, perpendicular to the main longitudinal veins of the two identical wings[6], would have required important longitudinal movements of the tegmina, which is nearly impossible, unless they were rub on other parts of the body, such as the body sclerites or the legs. Moreover, the tegmina of Titanoptera were probably lying flat on the body like in roaches, as it is visible in *Gigatitan vulgaris*[6], and they did not have part of their surface folded laterally at right angle, as in extant Orthoptera. Some extant other polyneopteran insects can stridulate when the wings are oriented horizontally: Some mantids stridulate bending their abdomen against their hind wings[40,41]—but Titanoptera hind wings are devoid of specialized structures (Supplementary Fig. 5)—; some cockroaches stridulate using specialized structures of the forewing and of the pronotum[42]—structures which are absent in Titanoptera. In addition, the specialized structures of Titanoptera are in the middle of the wings (and not on the basal or proximal parts), making the rubbing of these structures against the pronotum impossible. Thus, it is unlikely that Titanoptera could have stridulated as some extant mantids or cockroaches do.

The castanet hypothesis would imply the presence of a beater and of broadened areas on Titanoptera hind wings, structures that were never observed on the rare known hind wings of these insects[6]. As for a castanet mechanism involving only forewings, it is improbable due to the horizontal position of the wings in Titanoptera (unless the species lift their wings above their body to hit them together). Hence, this hypothesis seems unlikely.

The enlarged areas with simple large cells of the forewings of *Theiatitan*, *Paratitan* Sharov, 1968, or *Minititan* are very similar to those observed in the crepitating Acridoidea. But the reduced size of the vannus of their hind wings indicates that almost all well-preserved Titanoptera were probably poor flyers. A stationary crepitation hypothesis remains an option but it is unlikely

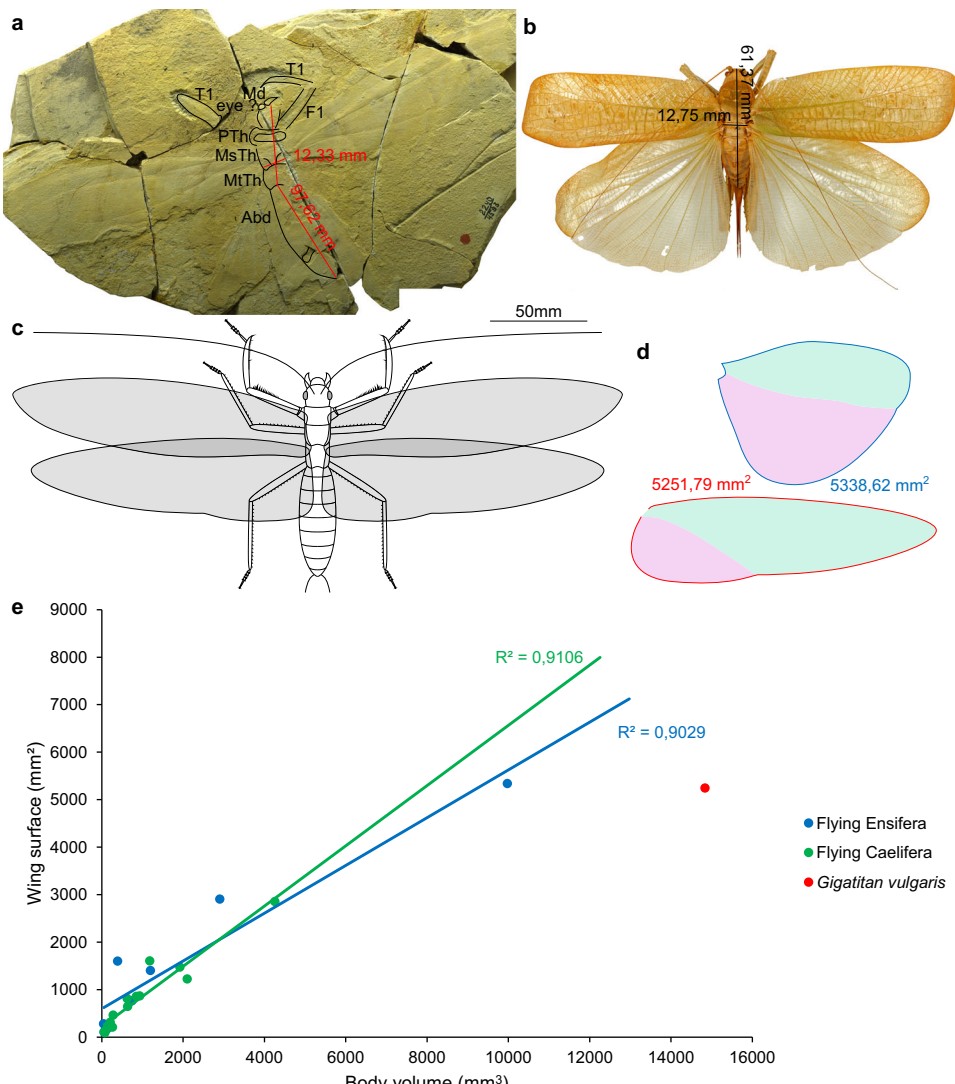

**Fig. 6 Flying ability of *Gigatitan vulgaris*. a** Specimen PIN 2240/4593; PTh prothorax, MsTh mesothorax, MtTh metathorax, Abd abdomen, Md mandible, F1 profemora, T1 protibia. **b** *Pseudophyllanax imperialis*, one of largest extant flying Orthoptera. **c** Reconstruction of *Gigatitan vulgaris*. **d** Hind wings of *Pseudophyllanax imperialis* (top, from (**b**)) and *Gigatitan vulgaris* (bottom, after[6]) with areas; note the differences in size of the vannus (in pink). **e** Wing surface plotted against body volume for several extant flying Orthoptera and *Gigatitan vulgaris*. *Gigatitan* seems too heavy to be able to fly actively. **a–d**: same scales.

in the Titanoptera with large cells subdivided into numerous smaller ones such as *Clatrotitan*, as this structure must limit the elasticity of the forewing membranes and so their capacity to crepitate.

A last, non-exclusive, hypothesis involves visual communication: some Titanoptera could have generated light flashes with their fenestrated forewings. These would have been produced when the insect moved its forewings, like many butterflies do during flight but also at rest. In the 'specialized' apparatus of *Clatrotitan*, each specialized large cell is subdivided into a couple of two flat surfaces of different sizes and orientations (Fig. 4) that would have allowed light reflections in a privileged direction, according to two hypotheses. In the first hypothesis, the emergent light ray results from a successive reflection of light by each couple of surfaces. This combined light ray has a stable emergent angle relatively to the incident light (see Supplementary Discussion: Reflection of light; Supplementary Fig. 6; Supplementary Tables 1–2; Supplementary Movies 1–2). This doubly reflected light is obliterated by the direct emergent light at the end of the

movement of the two surfaces. If the insect quickly moves its wings, this can generate a visual signal in a stable direction. However, this hypothesis does not explain why the two cells in each couple differ in color (for a same function), need a precise configuration (only working well for less than a third of the possible incident angles, thus preventing a general use to disturb predators), and give relatively low intensity flashes (about a third of the maximum simple reflection optimal configuration). The angle between the two cells of a couple is far from optimal to produce flashes by another way: a more acute angle would have allowed the intensity of the flashes and the incident angles to be increased, where a double reflection is possible.

In the second hypothesis, the light is directly reflected on one of the two cells of a couple. The smaller cells are strongly inclined, which allow light to be reflected in a very different angle from the angle of the rest of the wing, leading to a high contrast flash (up to three times higher than the double-reflection one). If the insect quickly moves its wings, this will generate a visual signal only for precise angles of the wing,

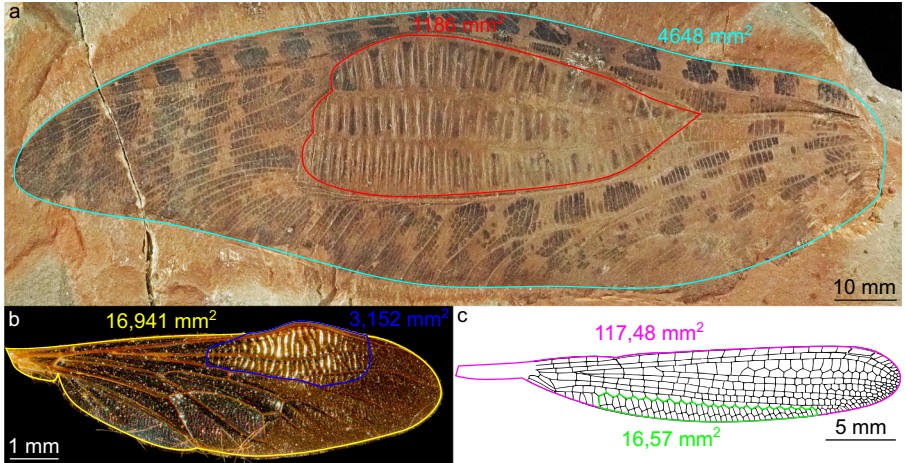

**Fig. 7 Surfaces of putative wing reflectors in several insects. a** *Clatrotitan andersoni.* **b** *Ommatius torulosus.* **c** *Aulliella crucigera.* The specialized zone of *Clatrotitan* is proportionally larger that in others insects.

leading to a very dynamic situation similar to what is known in modern insects using flashes. Those flashes could be achieved under almost all angles between the observer, light source and the insect, then is really simple to perform and allows an efficient use against predators.

In both hypotheses, the addition of all the resulting parallel light rays coming from the important series (ca. 80 specialized cells in *Clatrotitan*) of cell couples gives important flashes of light: this results in a 12 cm² specialized zone, covering a quarter of the whole wing, more than in other modern insects with corrugated zones (18% of the wing in *Ommatius torulosus* (Becker, 1925); 14% in the steleopterid *Aulliella crucigera* Pritykina, 1968 (see Fig. 7), and allowing to reflect the full solar light to an observer situated at a distance of up to a few meters. The light flashes hypothesis does not explain the whole variability of titanopteran broadened areas, but is supported by a body of evidence for at least some of them.

The flashes and crepitation hypotheses—best-supported—could be indirectly tested, looking for the receptive sensory structures in the Titanoptera: compound eyes and ocelli to detect flash signals (see Supplementary Fig. 1), versus specialized setae to detect air waves and/or tympanal organs to detect variations in air pressure. Unfortunately, these structures remain unknown in the Titanoptera, even though eyes and ommatidia are more probable than tympana, as they are plesiomorphic in Hexapoda. Nevertheless, the flash producing mechanism is more likely for *Clatrotitan* while the crepitation could be preferred for *Paratitan*, *Minititan*, or *Theiatitan*. Taxa like *Nanotitan* or *Gigatitan vulgaris* could have had a 'mixed' apparatus able to crepitate and produce flashes of light (see Supplementary Discussion: Putative modes of communication among Titanoptera). Like the numerous extant orders of arthropods which are known to rely on several sensorial modalities[43], Titanoptera could have been able to communicate using several modalities, increasing their significance for evolutionary biology.

If it is clear that the specialized structures in Titanoptera were used for communication, it is too early to ascertain whether it was for intra- or interspecific interactions. The two hypotheses are not exclusive: Both crepitations and light flashes could have been used to escape predators, attract sexual partners, or even in male–male competition through territory signaling for example. In many animals, males and females communicate before mating. For predators like the Titanoptera, communicating to gain insight on female receptivity and avoid being considered as a potential prey

could even have been vital. Then, the putative presence of the same structures in males and females do not eliminate the pre-mating communication hypothesis. The predation hypothesis cannot be ruled out either because putative terrestrial vertebrate predators of Titanoptera certainly had eyes so that flashes of light would have likely been an efficient defense strategy. In the same way, it is not possible to determine if the Titanoptera were diurnal or nocturnal insects (Supplementary Discussion: Titanoptera as diurnal insects). But both diurnal and nocturnal insects can have visual defenses against predators, based on structures and behaviors that work only during day, as exemplified by the wing color patterns of the nocturnal Saturniidae efficient against diurnal birds[44]. *Clatrotitan andersoni* and *Gigatitan vulgaris* could produce flashes and had disruptive color patterns on their tegmina, with alternate dark and clear bands of colors. Experiments have demonstrated that flash behaviors increase the survival of otherwise cryptic insects[45]. Flashes moreover tend to be more effective in large putative preys (as Titanoptera) and only a conspicuous flash display can substantially reduce predation pressure[29].

Sounds as a defense mechanism might seem less supported because the presence of ears is controversial for the Late Carboniferous tetrapods[46]. Still, some of them may have had the capacity to pick up sounds or ground vibrations[47] so that producing sudden sounds would also have been an effective way to startle or deter predators.

The relatively small archaeorthopteran order Titanoptera is remarkable in many ways, such as their large size and predatory legs. They had also very strange specialized zones on their forewings, formerly considered as 'resonators' of stridulatory apparatuses, which is unlikely. These structures were highly diverse across the order and could have allowed Titanoptera to communicate, most probably through crepitation and/or production of light flashes either during flight or at rest (Fig. 5e). Our discovery of a Carboniferous Titanoptera shows that sound and/or light communication is a very old phenomenon, ca. 50 Ma older than previous records as the oldest stridulating 'orthopteroids' are Capitanian (265.1 ± 0.4−259.9 ± 0.4 Ma), while *Theiatitan* is Moscovian (307.0–315.2 Ma; Fig. 8). These types of communication with the wings have evolved numerous times from non-homologous structures, in Diptera[48], Lepidoptera, Odonatoptera, and Titanoptera (for wing-based light communication), and in Lepidoptera, 'orthopteroid' orders Titanoptera, Caloneurodea, and subgroups of Orthoptera[49] (for wing-based

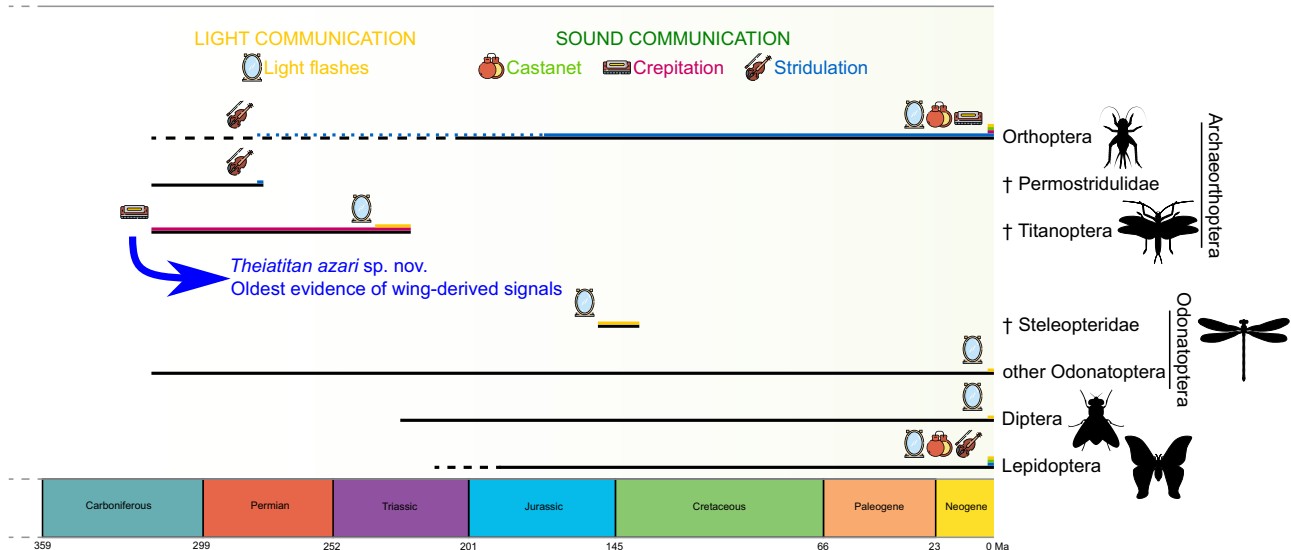

**Fig. 8 Insect lineages using wings to produce light and/or sound.** *Theiatitan azari* Schubnel, Roques & Nel, sp. nov. featured as oldest potential record of such a production. Only lineages with extinct species assumed to have produced sound or light with their wings are depicted, together with extant lineages that use those communication channels. Buzzing and lineages communicating with other structures, e.g., cicadas or fireflies, are not considered. Ages of lineages derived from fossil records; dotted lines: stem-lineages. † symbolizes extinct taxa. Permian stridulating Orthoptera correspond to new, undescribed Chinese Ensifera (Huang et al., in prep.). Harmonica, mirror and violin icons from Open Access 'Freepik @flaticon', castanet icon modified from Open Access 'Good Ware @flaticon'.

sound communication). Those multiple origins suggest that wing-based communication has been an important and innovative factor in the establishment of the deep past biodiversity, especially in the whole superorder Archaeorthoptera and its extant subclades[50].

## Methods

**Localities and repositories.** Moscovian (Westphalian C/D or equivalent Bolsovian/Asturian), Carboniferous, 'Terril N 7', containing rocks from the slag heap of coal mines 3 and 4 of Liévin, Avion, Pas-de-Calais, France. Repository. Muséum national d'Histoire naturelle, Paris (MNHN).

**Morphological observations.** The holotype of *Theiatitan azari* was studied with stereo microscope NIKON SMZ 1500 and SMZ 25. Microphotographs were made with digital cameras, and a focus stacking software Helicon Focus TM was used to increase the depth of field. All images were processed with Adobe Photoshop CS. A principal component analysis (PCA) was performed to visualize and quantify the slight relief and angularity of the fossils' wing structures associated to depth map. A more detailed account of Materials and methods is available online in the Supporting Information (S1 Text).

**Nomenclatural acts.** This published work and the nomenclatural acts it contains have been registered in ZooBank, the proposed online registration system for the International Code of Zoological Nomenclature (ICZN). The ZooBank LSIDs (Life Science Identifiers) can be resolved and the associated information viewed through and standard web browser by appending the LSID to the prefix "http://zoobank.org/". The LSIDs for this publication are urn:lsid:zoobank.org:pub:5F33D135-D38F-4D10-ADF5-9CE9BBFDC4C4.

**Reporting summary.** Further information on research design is available in the Nature Research Reporting Summary linked to this article.

## Data availability

All data needed to evaluate the conclusions are present in the paper and/or the Supplementary Information files. All data related to this paper are available from the corresponding authors upon reasonable request. No ethical approval or guidance were necessary.

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

# ARTICLE

19. Riede, K. A comparative study of mating behaviour in some Neotropical grasshoppers (Acridoidea). *Ethology* **76**, 265–296 (1987).
20. Gerhardt, H. C. & Huber, F. *Acoustic Communication in Insects and Anurans* (University of Chicago Press, 2002).
21. Surlykke, A. & Gogala, M. Stridulation and hearing in the noctuid moth *Thecophora fovea* (Tr.). *J. Comp. Physiol. (A)* **159**, 267–273 (1986).
22. Otte, D. A comparative study of communicative behavior in grasshoppers. *Miscellaneous Publ. Mus. Zool. Univ. Mich.* **141**, 1–168 (1970).
23. Otte, D. T*he North American Grasshoppers. Vol. 1. Acrididae: Gomphocerinae and Acridinae.* (Harvard University Press, 1981).
24. Lorier, E. Acoustic behaviour of *Metaleptea adspersa* (Orthoptera: Acrididae). *Can. Entomologist* **134**, 113–123 (2002).
25. Sanborn, A. F. & Phillips, P. K. Analysis of acoustic signals produced by the cicada *Platypedia putnami* variety *lutea* (Homoptera: Tibicinidae). *Ann. Entomological Soc. Am.* **92**, 451–455 (1999).
26. Lorier, E., Clemente, M. E., García, M. D. & Presa, J. J. El comportamiento acústico de *Fenestra bohlsii* Giglio-Tos (Orthoptera: Acrididae: Gomphocerinae). *Neotropical Entomol.* **39**, 839–853 (2010).
27. Smith, G. S. Structural color of *Morpho* butterflies. *Am. J. Phys.* **77**, 1010 (2009).
28. Niu, S.-C. et al. Angle-dependent discoloration structures in wing scales of *Morpho menelaus* butterfly. *Sci. China Technol. Sci.* **59**, 749–755 (2016).
29. Bae, S., Kim, D., Sherratt, T. N., Caro, T. & Kang, C. How size and conspicuousness affect the efficacy of flash coloration. *Behav. Ecol.* **30**, 697–702 (2019).
30. Lavigne, R. J. Evolution of courtship behaviour among the Asilidae (Diptera), with a review of courtship and mating. *Stud. Dipterologica* **9**, 703–742 (2002).
31. Lima, S., Vieira, R., Camargo, A. & Chagas, C. *Ommatius*: synonyms, new record, redescription of *Ommatius erythropus* and description of the female of *Ommatius trifidus* (Diptera: Asilidae: Ommatiinae). *Zoologia* **34**, 1–11 (2017).
32. Wootton, R. J. The geometry and mechanics of insect wing deformations in flight: a modelling approach. *Insects* **11**, 1–19 (2020).
33. Fleck, G., Nel, A., Bechly, G. & Martínez-Delclòs, X. Revision and phylogenetic affinities of the Jurassic Steleopteridae Handlirsch, 1906 (Insecta: Odonata: Zygoptera). *Insect Syst. Evolution* **32**, 285–305 (2001).
34. Zheng, D.-R., Nel, A. & Jarzembowski, E. A. The first Cretaceous damselfly of the Jurassic family Steleopteridae (Odonata: Zygoptera), from Surrey, England. *Cretac. Res.* **93**, 1–3 (2019).
35. Groover, R. S. Do dragonflies respond to sound? *Argia* **29**, 12–13 (2017).
36. Gorochov, A. V. New and little known Mesotitanidae and Paratitanidae (Titanoptera) from the Triassic of Kyrgyzstan. *Paleontological J.* **37**, 400–406 (2003).
37. Grimaldi, D. A. & Engel, M. S. *Evolution of the Insects*. (Cambridge University Press, 2005).
38. Stylman, M., Penz, C. M. & DeVries, P. Large hind wings enhance gliding performance in ground effect in a Neotropical butterfly (Lepidoptera: Nymphalidae). *Ann. Entomological Soc. Am.* **113**, 15–22 (2020).
39. Suarez-Tovar, C. M. & Sarmiento, C. E. Beyond the wing planform: morphological differentiation between migratory and non-migratory dragonfly species. *J. Evolut. Biol.* **29**, 690–703 (2016).
40. Gemeno, C., Claramunt, J. & Dasca, J. Nocturnal calling behavior in mantids. *J. Insect Behav.* **18**, 2005) 403–389.
41. Hill, S. A. Sound generation in *Mantis religiosa* (Mantodea: mantidae): stridulatory structures and acoustic signal. *J. Orthoptera Res.* **16**, 35–49 (2007).
42. Hartman, H. & Roth, L. Stridulation by the cockroach *Nauphoeta cinerea* during courtship behavior. *J. Insect Physiol.* **13**, 579–582 (1967).
43. Choi, N. et al. A mismatch between signal transmission efficacy and mating success calls into question the function of complex signals. *Anim. Behav.* **158**, 77–88 (2019).
44. Collins, M. M. Interpretation of wing pattern elements in relation to bird predation on adult *Hyalophora* (Saturniidae). *J. Lepidopterists' Soc.* **67**, 49–55 (2013).
45. Loeffler-Henry, K., Kang, C.-k, Yip, Y., Caro, T. & Sherratta, T. N. Flash behavior increases prey survival. *Behav. Ecol.* **29**, 528–533 (2018).
46. Müller, J., Bickelmann, C. & Sobral, G. The evolution and fossil history of sensory perception in amniote vertebrates. *Annu. Rev. Earth Planet. Sci.* **46**, 495–519 (2018).
47. Clack, J. A. & Anderson, J. S. Chapter 4. In *Evolution of the Vertebrate Ear—Evidence from the Fossil Record* (eds. Clack, J. A. et al.) Springer, *Handbook of Auditory Research* https://doi.org/10.1007/978-3-319-46661-3_4 (2016).
48. White, T. E., Vogel-Ghibely, N. & Butterworth, N. J. Flies exploit predictable perspectives and backgrounds to enhance iridescent signal salience and mating success. *Am. Naturalist* **195**, 733–742 (2020).
49. Desutter-Grandcolas, L. et al. 3-D imaging reveals four extraordinary cases of convergent evolution of acoustic communication in crickets and allies (Insecta). *Sci. Rep.* **7**, 1–8 (2017).
50. Robillard, T. & Desutter-Grandcolas, L. Phylogeny of the cricket subfamily Eneopterinae (Orthoptera, Grylloidea, Eneopteridae) based on four molecular loci and morphology. *Mol. Phylogenet. Evol.* **40**, 643–661 (2006).

## Acknowledgements

We sincerely thank Mr. Stéphane Carlier, Eiffage Route Nord Est, for the kind authorization to collect fossil insects in the terril of Avion. We thank Claire Mellish for her kind help in the study of the material of the NHM. We also thank Patrick Smith and Matthew McCurry (Australian Museum of Natural History) for the photograph of *Clatrotitan andersoni* (AMF36274), Pr. Alexander Rasnitsyn (Academy of Sciences, Moscow), Dmitry Shcherbakov and Dmitri Vasilenko for the photographs of *Mesotitanodes similis* (Sharov, 1968) (PIN 2785/2029), *Gigatitan vulgaris* Sharov, 1968 (PIN 2555/1541 and PIN 2240/4593). We thank Ben Warren (MNHN, Paris) for his proofreading of the manuscript.

## Author contributions

L.D.-G. and A.N. designed the research. P.R. collected and prepared the type specimen. T.S., R.G., R.C., M.P., and A.N. performed morphological, comparative studies, and line drawings. T.S., R.G., and R.C. took the photographs and prepared the life reconstruction artwork. T.S., F.L., L.D.-G., R.G., and A.N. drafted the paper.

## Competing interests

The authors declare no competing interests.
