## [Transparent Peer Review File · Communications Biology]

Reviewers' comments:

Reviewer #1 (Remarks to the Author):

In this manuscript, authors describe a new fossil Titanoptera species from the Carboniferous. In describing the fossil, the authors speculate on the function of the broadened zones in the forewing and offer interesting hypotheses. Previously, the function of these structures was considered a part of stridulatory apparatus (resonator) despite the fact that stridulatory files and a scraper have never been described from Titanoptera. Based on comparison with extant insect groups, the authors propose that the function could have been either to produce sound in a different manner (crepitation, castanet) or not related to acoustic signaling at all (light communication). The speculations are compelling and worth publication, but they are nonetheless speculations without much evidence. The issue of attributing a function to a structure in an extinct insect species is that it is difficult to test this hypothesis. I have a suggestion on how the authors can test their hypotheses, but as of now, the manuscript reads as an interesting evolutionary "story" rather than rigorous hypothesis testing. In making claims, I would like the authors to be more cautious.

Titanopterans were large carnivorous insects that had raptorial forelegs and no jumping legs. Not much is known about the specific morphological structures found in other body parts other than the forewings. They have highly specialized tegmina with unique broad zones in the middle. These zones were assumed to function as resonators of male-specific stridulatory apparatus. If there is sexual dimorphism in some sort of signaling device, we can safely infer that the device would have been used for sexual communication. However, both males and females of *Gigatitan vulgaris* apparently had these structures, according to Sharov. If this is true, then the function of these structures is not necessarily for sexual communication, but for some other communication, such as defensive signaling or something else. In fact, I have not really seen much evidence that the structure is male-specific or not. This inference probably came from the fact that modern Ensifera have male-specific stridulatory apparatus, but that does not necessarily mean that the Carboniferous archaeorthopteran lineage also had a male-specific signaling device. In some ensiferan lineages, such as Anostostomatidae and Gryllacrididae, abdominal stridulatory apparatus are found in both sexes. In other words, without much evidence on whether the structure is male-specific or present in both sexes, it is difficult to go on about speculating the function. The authors considered the new fossil wing to be a male specimen, presumably because the wing has the unique zones, which is presumably found only in males. This is a circular logic. Without additionally associated fossilized structures suggesting the sex of the specimen (such as abdomen terminalia), and without strong evidence that this structure was only found in males (which is questionable), then the best course of action is to declare as sex unknown. This is important to address.

Now, the authors put forth four possibilities regarding the function of these unique zones in the wing: stridulation, crepitation, castanet, and light flashes. These are all interesting hypotheses, but there is no direct way to test these. And I think that the authors need to think a bit more about each hypothesis. Without access to the whole body fossil of Titanoptera, the inference must be based on the structures found in the extant insects.

1. Stridulation found in extant Orthoptera relies on a file and a plectrum, and the authors are correct in stating that there is no direct evidence of structures on the titanopteran wings suggesting stridulation. However, it is possible that the actual stridulatory structures are found in other body parts for which we do not have fossil evidence for. For example, without clear fossilized midlegs or hind legs of the titanopterans, which could have included a scraper (or plectrum), it is difficult to entirely dismiss the stridulation idea. The authors make a statement: "stridulation is possible only because at least the lateral part of the wing is oriented vertically." This is only true in Orthoptera, but other polyneopteran groups can stridulate when the wings are oriented horizontally (mantids and roaches). In other words, I have not found any strong evidence to completely discount stridulation as a possibility. If we think about the possibility of this large insect using wings to stridulate to produce

defensive signaling, I think that it could be a very effective way to deter predators.

Some helpful citations:

Hill, S. A. (2007) Sound generation in *Mantis religiosa* (Mantodea: Mantidae): stridulatory structures and acoustic signal. *Journal of Orthoptera Research*, 16, 35-49.

HARTMAN, H., ROTH, L. Stridulation by a Cockroach during Courtship Behaviour. *Nature* 213, 1243-1244 (1967).

2. Crepitation is a possibility. It's most well-known in Oedipodinae, but also found in Acridinae within Acrididae. However, in grasshoppers, crepitation works only during flight, which means that it is functionally only possible in strong fliers. Plus, grasshoppers use their hindwings to crepitate, not forewings. One example that the authors did not cite (but cited the paper nonetheless) is found in Riede 1987 showing the hindwing-resonator of *Hyalopterix rufipennis*, which superficially resemble the titanopteran forewings. In this species, a clear, bell-like tone with a carrier frequency of 3kHz is produced by the flapping of the wings without stridulation. So perhaps, this is something to consider. Titanopterans were much larger than oedipodine grasshoppers. It is difficult to assess flight capacity of titanopterans, given the large wingspan and the overall shape of the tegmina, I suspect that these insects were not very strong fliers. I would strongly suggest that the authors include a discussion about the body size and scale in comparison to the extant species that use crepitation. One example use (and featured in the figure 2b) is *Stauroderus scalaris*, which actually uses the broad zones in the forewing for stridulation using hind legs rather than for crepitation (<https://youtu.be/Q47ETuuVBUY>).

3. Castanet idea is also a possibility. However, this also relies on the premise that the titanopterans beat their large wings up and down fast enough to create sound. In terms of biomechanics, it makes sense to have such structure at the edge of the wings (like what is found in the grasshoppers), rather than the central location in the titanopterans.

4. Light flashes. This is a novel idea, and pretty interesting, but it relies on a few premises. First, it requires that the titanopterans were diurnal and had a very strong visual sensitivity to flash patterns. We do not have much evidence for or against it. Second, given the examples that the authors used (damselflies, butterflies, flies), it requires that the titanopterans were exceptionally strong fliers for the light signal to be an effective means for communication. This is also based on the premise that this signaling was for mate recognition and both sexes were mostly flied. When we look at extant mantids, which are probably most similar to titanopterans, they are not necessarily known for strong flight capacity. Third, it requires that the titanopteran wings were highly transparent and reflective. Most of the extant orthopteran tegmina are colored and not reflective. The authors make a case that the corrugated pattern in the titanopteran wings could allow light to reflect and showed an example of asilid fly wing (Figure 2e). The corrugated pattern could be to strengthen the membranous part of the wing, and may have nothing to do with light reflection. So, as much as I would like to believe that the Carboniferous top insect predators use light flashes for communication (which would be spectacular), I just don't think that there is enough evidence for it. One possibility is to create a 3D print of the forewing using clear plastic, and actually test the idea of light reflection. Then, I will believe it.

Some of the titanopterans have very distinct specialized zones as shown in Figure 5, there is a lot of variation across species, as shown in Figure 4. The fact that they are found consistently across species indicates that it has a clear function, and that the fact that there is variation across means that they are under selection. The authors have proposed various hypotheses, but unfortunately not one is more convincing than others.

One important finding of this manuscript is the age of the new fossil, which now represents the oldest known Titanoptera. If titanopterans indeed used acoustic signaling, that would indicate that the Carboniferous was a really exciting period of the evolution of communication modalities.

Reviewer #2 (Remarks to the Author):

This is an interesting and thought-provoking paper on a new theory on the function of the modified areas of Titanoptera wings. The author's suggestion that they could be for visual communication rather than creating sound is novel and would be of interest to the community. Their argument is well thought through and I do not have any criticisms on this part of the paper which deserves to be published.

However, one part of the paper which I categorically reject is the attribution of the new taxon *Theiatitan azari* to the Titanoptera. This taxon is based only on one fragmentary wing and is open to a different interpretation.

In the supplementary information the authors state that it belongs to the 'Archaeorthoptera because of the basal fusion of R, M and CuA' however this is characteristic of Hemiptera. They also record veinlets between C and ScP and emerging from ScP. Where? These cannot be seen on the photo and have not been drawn on the line drawing. Besides Hemiptera (such as Fulgoromorpha) can also have these, as well as cross-veins and a long CuA parallel to M. Therefore I do not accept that this new taxon belongs to the Titanoptera.

For this reason I am suggesting a major revision with the new theory published in it's own right and the section on the new taxon and all mentions of the new taxon being deleted entirely, as it is flawed and therefore does not support their theory.

Reviewer #3 (Remarks to the Author):

This is a clearly-written and amply-supported manuscript on the early evolution of acoustic communication in insects. The authors address the claim, cited in the literature since 1937, that sound emission by wing stridulation appeared in an ancient orthopteroid lineage (Titanoptera), now extinct, during the Carboniferous (310 mya). By careful analysis of fossil specimens and focusing on the wing structures claimed to represent a stridulatory apparatus, the authors consider the several different mechanisms of sound production known from extant insects. None of these mechanisms is strongly upheld by examining the fossil structures. Rather, the structures – if they were involved in communication at all – may have been used for creating a moving, visual display. Because the Titanoptera were fully winged and presumably flew, it would be assumed that they had functioning compound eyes and vision capable of perceived optical signals. Alternatively, these signals – if they indeed existed – may have been intended for natural enemies. These inferences are then consistent with a much later origin for insect sound communication (late Palaeozoic), one for which there is rather solid evidence.

The authors could improve their manuscript in several ways. First, they might follow the inferences and possibilities indicated in the end of the above paragraph. That is, there is no firm evidence that the 'curious' wing structures were used generating visual signals, and the notion needs to be treated as a hypothesis, not as a definitive finding of the study (lines 19-20 of the abstract). Second, in assessing the sound emission hypothesis, it would be useful to consider the 'auditory environment' of the Carboniferous. Some of the sound producing mechanisms, e.g. crepitation, are typically used to deter natural enemies in extant insects, and they would not demand tympanic hearing in the Titanoptera (for which available fossil specimens do not provide any evidence for or against) : Is hearing known from any terrestrial animals of this period that might have been potential predators ? These lines of inquiry could make the article of greater interest to a wide readership.

Rebuttal letter for manuscript COMMSBIO-20-2602-T

Dear Editors and reviewers,

We thank you for giving us the opportunity to revise and resubmit our study. We also thank the three reviewers, who provided constructive comments to improve our paper. We have carefully taken into account all comments and suggestions. Among the most important changes, we have now revised the manuscript by addressing the following main points raised by the reviewers:

1) The flight capacities of the Titanoptera have been compared to those of the flying Orthoptera, resulting in the hypothesis that the Titans were not good flyers but could, at least glide. Crepitation can also be done without flying, like flashes of light.

2) A comparison has been done with mantids and cockroaches that stridulate, and we conclude that the morphology of the Titanoptera did not allow them to stridulate in the same way.

3) We provide additional evidence for the taxonomic attribution of the new fossil species to the Titanoptera (and not to the Hemiptera, in particular to the Auchenorrhyncha).

4) We provide additional evidence (including a mathematical demonstration and a short video) to critically assess the light flashes hypothesis.

5) We summarized evidence regarding the 'auditory environment' of the Carboniferous.

In the appended copy of the reply, we provide a point-by-point response to the reviewers' comments, with our responses in bold font. We hope this revised manuscript and the broad interest of our results will make a worthy contribution to **Communications Biology**.

Reviewer #1 (Remarks to the Author):

In this manuscript, authors describe a new fossil Titanoptera species from the Carboniferous. In describing the fossil, the authors speculate on the function of the broadened zones in the forewing and offer interesting hypotheses. Previously, the function of these structures was considered a part of stridulatory apparatus (resonator) despite the fact that stridulatory files and a scraper have never been described from Titanoptera. Based on comparison with extant insect groups, the authors propose that the function could have been either to produce sound in a different manner (crepitation, castanet) or not related to acoustic signaling at all (light communication). The speculations are compelling and worth publication, but they are nonetheless speculations without much evidence. The issue of attributing a function to a structure in an extinct insect species is that it is difficult to test this hypothesis. I have a suggestion on how the authors can test their hypotheses, but as of now, the manuscript reads as an interesting evolutionary "story" rather than rigorous hypothesis testing. In making claims, I would like the authors to be more cautious.

We acknowledge that some sentences should have been phrased more carefully and we have now modified the manuscript accordingly. We thank the reviewer for its suggestion

to test our hypotheses (see below) and appreciate that they find our results and speculations compelling and worthy of publication.

Titanopterans were large carnivorous insects that had raptorial forelegs and no jumping legs. Not much is known about the specific morphological structures found in other body parts other than the forewings. They have highly specialized tegmina with unique broad zones in the middle. These zones were assumed to function as resonators of male-specific stridulatory apparatus. If there is sexual dimorphism in some sort of signaling device, we can safely infer that the device would have been used for sexual communication. However, both males and females of *Gigatitan vulgaris* apparently had these structures, according to Sharov. If this is true, then the function of these structures is not necessarily for sexual communication, but for some other communication, such as defensive signaling or something else. In fact, I have not really seen much evidence that the structure is male-specific or not. This inference probably came from the fact that modern Ensifera have male-specific stridulatory apparatus, but that does not necessarily mean that the Carboniferous archaerotheran lineage also had male-specific signaling device. In some ensiferan lineages, such as Anostomatidae and Gryllacrididae, abdominal stridulatory apparatus are found in both sexes. In other words, without much evidence on whether the structure is male-specific or present in both sexes, it is difficult to go on about speculating the function. The authors considered the new fossil wing to be a male specimen, presumably because the wing has the unique zones, which is presumably found only in males. This is a circular logic. Without additionally associated fossilized structures suggesting the sex of the specimen (such as abdomen terminalia), and without strong evidence that this structure was only found in males (which is questionable), then the best course of action is to declare as sex unknown. This is important to address.

Yes, you are right, thanks for pointing it out. We have now declared the specimen sex as unknown to avoid circular reasoning.

Now, the authors put forth four possibilities regarding the function of these unique zones in the wing: stridulation, crepitation, castanet, and light flashes. These are all interesting hypotheses, but there is no direct way to test these. And I think that the authors need to think a bit more about each hypothesis. Without access to the whole body fossil of Titanoptera, the inference must be based on the structures found in the extant insects.

1. Stridulation found in extant Orthoptera relies on a file and a plectrum, and the authors are correct in stating that there is no direct evidence of structures on the titanopteran wings suggesting stridulation. However, it is possible that the actual stridulatory structures are found in other body parts for which we do not have fossil evidence for. For example, without clear fossilized midlegs or hind legs of the titanopterans, which could have included a scraper (or plectrum), it is difficult to entirely dismiss the stridulation idea. The authors make a statement: "stridulation is possible only because at least the lateral part of the wing is oriented vertically." This is only true in Orthoptera, but other polyneopteran groups can stridulate when the wings are oriented horizontally (mantids and roaches). In other words, I have not found any strong evidence to completely discount stridulation as a possibility. If we think about the possibility of this large insect using wings to stridulate to produce defensive signaling, I think that it could be a very effective way to deter predators.

Some helpful citations:

Hill, S. A. (2007) Sound generation in *Mantis religiosa* (Mantodea: Mantidae): stridulatory structures and acoustic signal. *Journal of Orthoptera Research*, 16, 35-49.

HARTMAN, H., ROTH, L. Stridulation by a Cockroach during Courtship Behaviour. Nature 213, 1243–1244 (1967).

We have elaborated on why we consider the stridulation hypothesis very unlikely (section ‘the nature of wing signaling in Titanoptera’). The reviewer is correct when mentioning polyneopteran groups that stridulate with horizontally oriented wings. Mantids stridulate using their hindwings, whereas cockroaches stridulate rubbing the proximal part of the forewings against a specific structure on the pronotum. The specialized structures of Titanoptera are found in forewings, which suggest that they were not able to stridulate like mantids; and Titanoptera do not have specialized structures like the one found on the pronotum of the roach *Nauphoeta cinerea*, here again suggesting that they did not stridulate like this cockroach. In addition, the specialized structures of Titanoptera are found in the middle of the wings (and not on the proximal part), which would prevent from the possibility to rub these structures against the pronotum. This is why we consider the stridulation hypothesis unlikely. Nonetheless, we have complemented the manuscript with those considerations to be as thorough as possible.

2. Crepitation is a possibility. It’s most well-known in Oedipodinae, but also found in Acridinae within Acrididae. However, in grasshoppers, crepitation works only during flight, which means that it is functionally only possible in strong fliers. Plus, grasshoppers use their hindwings to crepitate, not forewings. One example that the authors did not cite (but cited the paper nonetheless) is found in Riede 1987 showing the hindwing-resonator of *Hyalopterix rufipennis*, which superficially resemble the titanopteran forewings. In this species, a clear, bell-like tone with a carrier frequency of 3kHz is produced by the flapping of the wings without stridulation. So perhaps, this is something to consider. Titanopterans were much larger than oedipodine grasshoppers. It is difficult to assess flight capacity of titanopterans, given the large wingspan and the overall shape of the tegmina, I suspect that these insects were not very strong fliers. I would strongly suggest that the authors include a discussion about the body size and scale in comparison to the extant species that use crepitation. One example use (and featured in the figure 2b) is *Stauroderus scalaris*, which actually uses the broad zones in the forewing for stridulation using hind legs rather than for crepitation (<https://youtu.be/Q47ETuuVBUY>).

This is a good suggestion, thank you. We have included a discussion about flight capacity in titanopterans. We have produced an additional figure relating body size and wing surface with flight ability. Our results show that Titans were probably poor flyers, but also that some insects can crepitate without flying. Most of this additional work is provided in the Suppl. Mat.

Stauroderus scalaris is able to stridulate but is also well known for its crepitation. Several of us observed it *in natura*.

3. Castanet idea is also a possibility. However, this also relies on the premise that the titanopterans beat their large wings up and down fast enough to create sound. In terms of biomechanics, it makes sense to have such structure at the edge of the wings (like what is found in the grasshoppers), rather than the central location in the titanoptera.

This is true what in modern Orthoptera only the beater have to at the edge while the resonator could be more central, as figured in Figure 2.

4. Light flashes. This is a novel idea, and pretty interesting, but it relies on a few premises. First, it requires that the titanopterans were diurnal and had a very strong visual sensitivity to flash patterns.

Not necessarily. Many nocturnal insects have developed structures that are only visible by day, e.g. the ‘ocelli’ on the wings of many Attacidae, or in some Myrmeleontidae that are nocturnal. These structures have probably evolved as a way to confuse predators.

We do not have much evidence for or against it. Second, given the examples that the authors used (damselflies, butterflies, flies), it requires that the titanopterans were exceptionally strong fliers for the light signal to be an effective means for communication.

No, not necessarily. Movements of the forewings can cause flashes even when the insect is at rest. Individuals belonging to the butterfly genus *Morpho*, for instance, can produce light flashes in flight and at rest too.

This is also based on the premise that this signaling was for mate recognition and both sexes were mostly flied.

Not necessarily. These flashes can be also to disturb predators. Regarding intraspecific signaling, it happens in nature that both males and females produce signals before mating. More specifically, for predators like Titanoptera supposedly, male-female signaling could have been selected to assess the female receptivity before mating. We have elaborated on this issue in the manuscript and underlined that disentangling intra- vs interspecific communication hypotheses requires further evidence.

When we look at extant mantids, which are probably most similar to titanopterans, they are not necessarily known for strong flight capacity.

Mantodea are very far from Titanoptera, in phylogenetic sense, and also in wing morphology. They can hardly be used for comparison. But even many Mantodea can fly for some distance, and emitting songs or light flashes when flying even for a short distance could be possible for the Titanoptera that have not reduced wings, able to sustain them during short flights.

Third, it requires that the titanopteran wings were highly transparent and reflective.

Not necessarily. The male *Calopteryx* have blue metallic, non-transparent, wings that make flashes. It is also true in the genus *Morpho*.

Most of the extant orthopteran tegmina are colored and not reflective. The authors make a case that the corrugated pattern in the titanopteran wings could allow light to reflect and showed an example of asilid fly wing (Figure 2e). The corrugated pattern could be to strengthen the membranous part of the wing, and may have nothing to do with light reflection.

Right, we have given additional details to clarify this issue. In brief, the wing pattern in Titanoptera is better organized than a simply corrugated wing. In the case of the asilid, this structure is only in males, and so is certainly used in communication, and not only for strengthening the wing membrane as in the Scoliidae.

So, as much as I would like to believe that the Carboniferous top insect predators use light flashes for communication (which would be spectacular), I just don't think that there is enough evidence for it. One possibility is to create a 3D print of the forewing using clear plastic, and actually test the idea of light reflection. Then, I will believe it.

The reviewer raised three points, that we have now discussed in the manuscript, and made a suggestion that we have followed. More precisely, we have extended our discussion about the premises for light communication. We have also created a 3D print, as suggested, but its precision seems not sufficient to give good results. Then we created virtual simulation of the wing. Our results clearly indicate that light is reflected in *Clatrotitan* in precise directions. We have formalized the reflection through a theoretical model based on the pattern of the concerned cells and showed that the reflected light is 'accumulated' under precise incident angles. We have also done an experiment with mirrors and a source of parallel rays of light (as for sun) to show what can happen (with a short video as a supplementary file).

Some of the titanopterans have very distinct specialized zones as shown in Figure 5, there is a lot of variation across species, as shown in Figure 4. The fact that they are found consistently across species indicates that it has a clear function, and that the fact that there is variation across means that they are under selection. The authors have proposed various hypotheses, but unfortunately not one is more convincing than others.

Yes, we have clear evidence that this structure has a function and that they are under selection. Further evidence is still necessary to confirm some hypotheses or (definitely?) reject others. Still, we have collated as much evidence as possible and presented our arguments to underline the most likely functions. The penultimate paragraph of the discussion now summarizes this situation.

One important finding of this manuscript is the age of the new fossil, which now represents the oldest known Titanoptera. If titanopterans indeed used acoustic signaling, that would indicate that the Carboniferous was a really exciting period of the evolution of communication modalities.

Right, thanks for pointing it out. See also our answer to the last comment of reviewer #3.

Reviewer #2 (Remarks to the Author):

This is an interesting and thought-provoking paper on a new theory on the function of the modified areas of Titanoptera wings. The author's suggestion that they could be for visual communication rather than creating sound is novel and would be of interest to the community. Their argument is well thought through and I do not have any criticisms on this part of the paper which deserves to be published.

Thank you for this general comment and for reviewing our manuscript.

However, one part of the paper which I categorically reject is the attribution of the new taxon *Theiatitan azari* to the Titanoptera. This taxon is based only on one fragmentary wing and is open to a different interpretation.

In the supplementary information the authors state that it belongs to the 'Archaeorthoptera because of the basal fusion of R, M and CuA' however this is characteristic of Hemiptera.

They also record veinlets between C and ScP and emerging from ScP. Where? These cannot be seen on the photo and have not been drawn on the line drawing. Besides Hemiptera (such as Fulgoromorpha) can also have these, as well as cross-veins and a long CuA parallel to M. Therefore I do not accept that this new taxon belongs to the Titanoptera.

For this reason I am suggesting a major revision with the new theory published in it's own right and the section on the new taxon and all mentions of the new taxon being deleted entirely, as it is flawed and therefore does not support their theory.

Yes, Fulgoromorpha, but also more widely Auchenorrhyncha, but there are several crucial differences between their venations and that of our fossil.

Indeed, one of us has extensively studied the venation of the orthopteroids and of the Acercaria (including the Hemiptera) (citations of concerned papers added, these are now the bases for the venation studies on these groups)

We have re-observed the fossil and precised some points, and provided new photo and new drawing. Thanks for this remark, it allows us to clarify our rationale. We have thus explained why Acercaria are excluded, while there is no argument to exclude the Titanoptera but several favoring this attribution hypothesis.

Reviewer #3 (Remarks to the Author):

This is a clearly-written and amply-supported manuscript on the early evolution of acoustic communication in insects. The authors address the claim, cited in the literature since 1937, that sound emission by wing stridulation appeared in an ancient orthopteroid lineage (Titanoptera), now extinct, during the Carboniferous (310 mya). By careful analysis of fossil specimens and focusing on the wing structures claimed to represent a stridulatory apparatus, the authors consider the several different mechanisms of sound production known from extant insects. None of these mechanisms is strongly upheld by examining the fossil structures.

Rather, the structures – if they were involved in communication at all – may have been used for creating a moving, visual display. Because the Titanoptera were fully winged and presumably flew, it would be assumed that they had functioning compound eyes and vision capable of perceived optical signals. Alternatively, these signals – if they indeed existed – may have been intended for natural enemies. These inferences are then consistent with a much later origin for insect sound communication (late Palaeozoic), one for which there is rather solid evidence.

The authors could improve their manuscript in several ways. First, they might follow the inferences and possibilities indicated in the end of the above paragraph. That is, there is no firm evidence that the ‘curious’ wing structures were used generating visual signals, and the notion needs to be treated as a hypothesis, not as a definitive finding of the study (lines 19-20 of the abstract).

Right, we are now more cautious in our phrasings, as already pointed out by reviewer #1. We have clearly written that those are hypotheses and have discussed arguments and counter-arguments for each hypothesis.

Second, in assessing the sound emission hypothesis, it would be useful to consider the ‘auditory environment’ of the Carboniferous. Some of the sound producing mechanisms, e.g. crepitation, are typically used to deter natural enemies in extant insects, and they would not demand tympanic hearing in the Titanoptera (for which available fossil specimens do not provide any evidence for or against) : Is hearing known from any terrestrial animals of this

period that might have been potential predators ? These lines of inquiry could make the article of greater interest to a wide readership.

This is a nice suggestion, thank you. We have summarized what is known in this regard for putative vertebrate predators near the end of the manuscript.

REVIEWERS' COMMENTS:

Reviewer #1 (Remarks to the Author):

I have read the revised manuscript as well as the rebuttal, and the authors addressed the comments raised by the reviewers sufficiently. The additional evidence that the authors provided for the light flash hypothesis is interesting, but it is not clear what that demonstrates. I have watched the videos and read the supplementary materials. Based on the virtual simulation of the wing, the authors did show that the light would be reflected in precise directions, but did not take the scale into account. If the model is true to the size of the actual specimen, I am not sure if the effect would be the same. The experiments done with mirrors demonstrate the idea well, but if scaled down to the true size, would it actually sparkle? I am not sure. An added sentence regarding the scale and the limit of this experiment should be included.

The arguments discounting other more conventional acoustic communication methods are adequately presented.

In line 284, the authors write, "The predation hypothesis cannot be ruled out either because putative terrestrial vertebrate predators of Titanoptera certainly had eyes so that flashes of light would have likely been an efficient defense strategy." Would a vertebrate predator at that time be really deterred by sparkling light that these insects produced? Why? What makes the predators avoid the light reflection? There is no incentive to avoid, as titanopterans were probably not aposematic or had strong defensive mechanisms. I think that the light flash would have made the insects more noticeable. If the sole function of the light flash was for intraspecific communication, I would buy the argument, but I don't find the defensive function of this flash pattern as plausible. In order to make the light flash as a viable function, the authors need to make an assumption that the insects were diurnal. Please clearly state this assumption.

I feel like that the supplementary information contains a lot more interesting data than what is presented in the actual manuscript. The main manuscript reads a lot like speculation, whereas the supplementary data back up with some concrete data, so I suggest a bit of revision, bringing some parts of the supplementary data (such as Figure 6 and 7 or Table 2-3) to the main document.

Dear colleagues

We have carefully read and answered to the requests of the referee

In particular, we have transferred some text of the suppl. Mat. to the main text and added two figures to better explain the points and requests.

Thanks a lot for your patience and kind help

Pr Nel André

Answers to the referee

In blue

Reviewer #1 (Remarks to the Author):

I have read the revised manuscript as well as the rebuttal, and the authors addressed the comments raised by the reviewers sufficiently. The additional evidence that the authors provided for the light flash hypothesis is interesting, but it is not clear what that demonstrates. I have watched the videos and read the supplementary materials. Based on the virtual simulation of the wing, the authors did show that the light would be reflected in precise directions, but did not take the scale into account. If the model is true to the size of the actual specimen, I am not sure if the effect would be the same. The experiments done with mirrors demonstrate the idea well, but if scaled down to the true size, would it actually sparkle? I am not sure. An added sentence regarding the scale and the limit of this experiment should be included.

Thank you for your general comments.

We explored the effect of the size on the light flash hypothesis. The differences due to the size of the cell in the insect wing vs. the mirror has no impact on the 'behavior' of the light at these scales. It would be necessary to take into account the effect of the size in the case of extremely small structures, at the level of the micron.

It is impossible to give quantitative value of reflected light, as we do not know important characteristics of the wing such as its reflective index. Then we explored it through a comparative point of view. We found that the area of specialized wing zone of *Clatrotitan* is proportionally larger than in other light communicating insects, which supports a sufficient effect to be used in intraspecific communication. For an interspecific communication, we calculated that the specialized area of the wing is broad enough to reflect a high amount of light at several meters (even if we are not able to give a precise value of reflected light as explained earlier), making it an efficient anti-predation mechanism.

The arguments discounting other more conventional acoustic communication methods are adequately presented.

Thank you.

In line 284, the authors write, “The predation hypothesis cannot be ruled out either because putative terrestrial vertebrate predators of Titanoptera certainly had eyes so that flashes of light would have likely been an efficient defense strategy.”

Would a vertebrate predator at that time be really deterred by sparkling light that these insects produced? Why? What makes the predators avoid the light reflection? There is no incentive to avoid, as titanopterans were probably not aposematic or had strong defensive mechanisms. I think that the light flash would have made the insects more noticeable.

We have added several references explaining this mechanism. To put in a nutshell, light flashes could disorient a predator and temporally blinding it. Then the predator could even learn that light flashing preys are difficult to catch and avoid them afterwards.

If the sole function of the light flash was for intraspecific communication, I would buy the argument, but I don’t find the defensive function of this flash pattern as plausible. In order to make the light flash as a viable function, the authors need to make an assumption that the insects were diurnal. Please clearly state this assumption.

Unfortunately, we did not find a way to determine if the Titanoptera were diurnal or nocturnal insects. In our opinion they should have been diurnal as we explained in supplementary material. However both diurnal and nocturnal insects can have defenses against predators, based on structures and behaviors that works only during day, as demonstrated by the wing color patterns of the nocturnal Saturniidae, efficient against diurnal birds and lizards.

The same situation occurs with a nocturnal Neuroptera Myrmeleontidae that has eye spots on the wings.

I feel like that the supplementary information contains a lot more interesting data than what is presented in the actual manuscript. The main manuscript reads a lot like speculation, whereas the supplementary data back up with some concrete data, so I suggest a bit of revision, bringing some parts of the supplementary data (such as Figure 6 and 7 or Table 2-3) to the main document.

Thank you for this comment. We have tried to write the main text in such a way that it is easily understandable for a broad audience. That is why some important but technical points were only presented in the supplementary material. But we agree that some important points such as those related to the flight of the Titanoptera would benefit for additional explanations in the main text; so we have moved this part and the related figure in the main text.